# Energy Efficiency and Thermal Performance of Office Buildings Integrated with Passive Strategies in Coastal Regions of Humid and Hot Tropical Climates in Madagascar

**Modeste Kameni Nematchoua [1,2,3,*], Jean Christophe Vanona [4] and José A. Orosa [5]**

[1] Beneficiary of an AXA Research Fund Postdoctoral Grant, Research Leaders Fellowships, AXA SA 25 avenue Matignon, 75008 Paris, France

[2] Local Environment Management & Analysis (LEMA), Département d'Architecture, Géologie, Environnement et Construction (ArGEnCo), University of Liège, Allée de la Découverte 9, Quartier Polytech 1, 4000 Liège, Belgium

[3] Indoor Environmental Quality Lab, School of Architecture, Design and Planning, The University of Sydney, Sydney, NSW 2006, Australia

[4] Department of Civil Engineering, Polytechnic Higher School, University of Antsiranana, Antsiranana 201, Madagascar; christohpe@gmail.com

[5] Department of Energy and M. P. Escuela Técnica Superior de N. y M, University of A Coruña, Paseo de Ronda 51, 15011 A Coruña, Spain; jose.antonio.orosa@udc.es

* Correspondence: mkameni@uliege.be or modeste.kameni@sydney.edu.au or kameni.modeste@yahoo.fr; Tel.: +32-4-65178927 or +261322430984 or +61-421-600-048

**Abstract:** Researchers have used passive strategies, such as the implementation of thermal insulation and the use of phase change materials (PCM), in several studies, but some problems have not yet been solved. It is the case of showing the real effect of external shading combined with thermal insulation and phase change materials to improve the thermal performance and energy efficiency of office buildings in tropical coastal areas. Another pending problem to be solved is to define the impact produced by passive strategies on the performance of workers in office buildings in coastal zones. It is with a view to answering all these questions that this study was envisaged with the main objective of evaluating, analyzing, comparing, and discussing the effect of thermal insulation and phase change materials on thermal comfort and energy demand in coastal areas of hot and humid tropical climates located in the island of Madagascar. In this sense, hourly climate data for the past 30 years have served as the basis for assessing environmental conditions of future climate. It was found that the PCMs have a more significant effect on the coastal zone of hot climates than humid tropical climates. The results of the statistical analyses showed that the application of passive strategies stabilizes indoor air temperatures to between 23 °C and 28 °C in the offices, which is the recommended comfort range in these regions. In the coastal regions of Madagascar, up to 30% of cooling energy is expected to be reduced by combining the introduction of thermal insulation and PCM materials.

**Keywords:** thermal performance; office buildings; passive strategies; coastal regions; tropical climates

## 1. Introduction

Climate change is a source of upheaval in the usual way of life of many populations of the world [1]. The impact of this phenomenon, particularly drought and floods, affects several countries around the world [2]. Thus, for several decades, various studies have shown a sustained growth

in energy consumption for heating and cooling in habitats [3,4]. Given the risks of a shortage of fossil fuel sources and taking into account global population growth, newbuildings must meet more requirements than usual [5,6]. It is, therefore, a question of proposing residential buildings capable of ensuring acceptable comfort with a minimum of energy consumed, or even zero [7]. In most countries of the world, buildings currently represent 40% of the total energy consumption and 36% of total $CO_2$ emissions [8]. Predicting the energy consumption of residential buildings is fundamental to propose strategies to improve their energy performance with the aim of saving energy and reducing the environmental impacts of the buildings. Despite this, it is known that the energy system of buildings is very complex due to the great difference between them [8].

The application of passive strategy techniques, such as the use of phase change materials, is a novel idea for improving indoor air [9]. Indeed, the performance of new materials makes it possible to better conduct numerous experiments in the laboratory or in places with harsh climatic conditions [10]. Nowadays, some studies can be simultaneously realized in residential buildings [11]. The integration of these materials in building walls, making it possible to obtain a significant inertia in a reduced volume and under a small temperature interval, gave rise to new solutions to the problem of passive solar heating [12]. The phase change material (PCM) represents a sustainable alternative to reducing energy consumption [12,13] and, at the same time, increases the thermal comfort of occupants [14]. By consequence, its incorporation to reduce the heating and cooling energy demand of buildings has aroused the particular interest of many scientists, as it makes it possible to store and release large amounts of energy in the form of heat during the material melting and solidification process [14].

Air conditioning is not the ideal solution for lowering the temperature in a building, because it further increases energy consumption and requires rigorous maintenance to ensure air quality [15]. One of the ways to reduce the energy needs of a building is, therefore, the design of an energy-efficient envelope, limiting losses and recovering passive inputs as much as possible [16]. During the summer period, heat waves and high temperatures force the vulnerable population to consume more energy [17]. To achieve these objectives, there are a number of basic principles, the most important of which are thermal insulation, inertia, and the use of solar gain [18–20]. Following the existence of many passive strategies, a compromise is noticed between cooling and heating performance. The choice of different passive strategies, which could have a better overall impact on energy performance, depends on the type and also on the operation of the building [21].

On one hand, as an example of the type of building, Raffaele et al. [22] showed that, in the Mediterranean climate, their passive strategy reduced to between 40% and 60% of relative humidity levels. Another example [23] assessed the energy performance of office buildings located in the Mediterranean coastal region during two seasons (winter and summer). The study was based on physical measurements, the analysis of questionnaires, and the introduction of passive strategies. The indoor comfort temperature found was 26.3 °C in summer and 21.6 °C in winter, varying between 3.1 °C and 6.4 °C compared to the average outdoor temperatures obtained during the working hours.

Guan [24] studied, in air-conditioned office buildings located in some Australian cities, the effect of internal charge densities on energy and thermal performance. The results showed that the problem of overheating could be completely avoided if the energy consumption in buildings was reduced to between 89 and 120 kWh/m$^2$ per year, with the total internal load density of the building also being reduced.

On the other hand, the performance of workers in their office varies according to indoor climatic conditions. Very often, buildings located in the coastal tropical zones are regularly uncomfortable because of their geographical position.

Finally, focusing on the studies found in the literature, it is normal to conclude that several works have studied the performance of passive strategies on thermal comfort and energy consumption in office buildings. However, the following identified problems are still unsolved.

(a) The energy efficiency and thermal performance of buildings incorporating thermal insulation and phase change materials (PCM) materials, located in coastal regions with a hot and humid tropical climate, have not yet been studied.

(b) The relationship between carbon reduction, energy savings, and weather factors has not yet been assessed in the tropical coastal zones.

In all regions of the world, coastal zones are highly exposed to the harmful effects of climate change, so it is very difficult to propose a common international standard relative to these regions. In fact, the different impacts vary from one region to another. The island of Madagascar is one of the regions of the world where biodiversity and buildings are strongly impacted by the seasonal variability of the climate. Thus far, no study or standard has been established in the big island to improve the performance of residential and commercial buildings. Little is known about the various passive strategies to the population of the island of Madagascar.

For the moment, no study exists in the literature aiming to improve the performance of workers in buildings located in the coastal zone in the six Indian Ocean countries. By consequence, the main purpose of this research is to evaluate, analyze, and compare the impact of passive strategies in office buildings located in the coastal area. This research was grouped in three main sections. The methodology section shows the different methodological details of this research; the results, its analysis, and comparison are shown in Section 3. The last section constitutes some of the main findings.

## 2. Methodology

The different stages followed in this study are described in Section 3.1 to Section 3.5.

### 2.1. Studied Locations

The Madagascar island is considered in terms of area and population to be the largest of the Indian Ocean islands, with an estimated area of 592,000 km$^2$. It is dominated by the tropical climate, which is divided into two seasons: A rainy season from November to April and a dry season from May to October. In particular, Madagascar stretches between 20°0′ South latitude and 47°0′ East longitude, covering different climatic zones in which four of the cities studied are located. All these cities were selected due to being located in the coastal zone, with significant solar and wind power potential, as a consequence of their geographic location, which is very favorable for tourism and for their cultural diversity and micro-climate. The local position of each studied city in this research is shown in Figure 1.

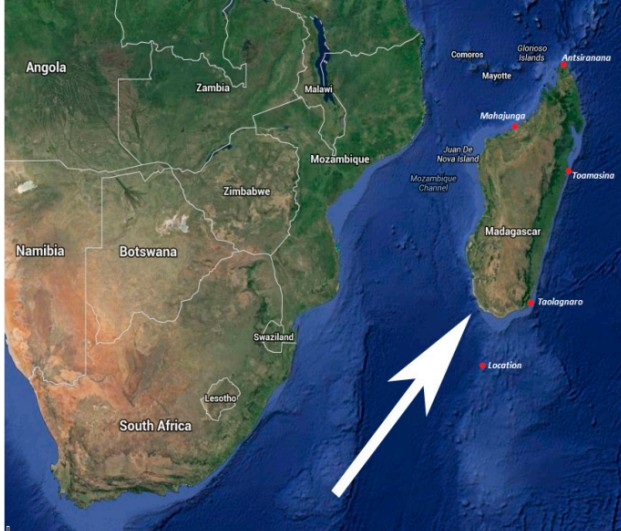

**Figure 1.** Madagascar map showing the four selected locations.

From this figure, it can be observed that the four studied cities are located in the coastal region, and the majority of descriptions is listed in Table 1. These data were provided in 2019.

**Table 1.** Climatic characteristics of four cities.

| Cities | Latitude | Longitude | Altitude (m) | Temp. (°C) | | RH (%) | | Wind Speed (m/s) | |
|---|---|---|---|---|---|---|---|---|---|
| | | | | Max. | Min. | Max | Min. | Max. | Min. |
| Antsiranana | 12.2S | 49.27E | 40 | 33.5 | 20.0 | 85.0 | 60.0 | 9.0 | 4.5 |
| Mahajunga | 15.53S | 46.29E | 20 | 37.5 | 15.0 | 90.0 | 40.0 | 8.5 | 0.1 |
| Toamasina | 18.19S | 49.34E | 06 | 35.5 | 17.0 | 80.0 | 50.0 | 8.5 | 0.0 |
| Taolagnaro | 25.1S | 46.69E | 539 | 30.0 | 17.0 | 90.0 | 17.0 | 5.6 | 4.2 |

In Table 1, it can be deduced that the four selected cities are located in a region with different climate.

### 2.2. Madagascar Climate Zones

In Madagascar, there are six main climatic zones [25]:

- The North region dominated by the tropical transition climate;
- The Northwest region dominated by a warm tropical climate;
- The Southwest region dominated by a sub-arid tropical climate;
- The South region dominated by an arid tropical climate;
- The Center region dominated by the tropical climate of altitude;
- The Northeast and Southeast regions, which are dominated by a humid tropical climate.

Antsiranana city is located in the North region of Madagascar dominated by the tropical transition climate; Mahajunga city is located in the Northwest region dominated by a warm tropical climate, and Taolagnaro and Toamasina cities are dominated by a humid tropical climate.

In this research, the different hourly outdoor data of air temperature, relative humidity, radiation, and wind speed between the others for the four studied cities were provided by Meteonorm software version 7.7.3 employing, as input data, the longitude, latitude, and altitude of each city. This tool allows us to download the weather data of any country under basic longitude and latitude.

### 2.3. Office Buildings

The building studied in this research consists of several administrative offices occupied by employees of the Ministry of Public Works of the city of Antsiranana for more than 20 years. The same office was built in the three other cities of the country with different climatic characteristics like that showed in Figure 2.

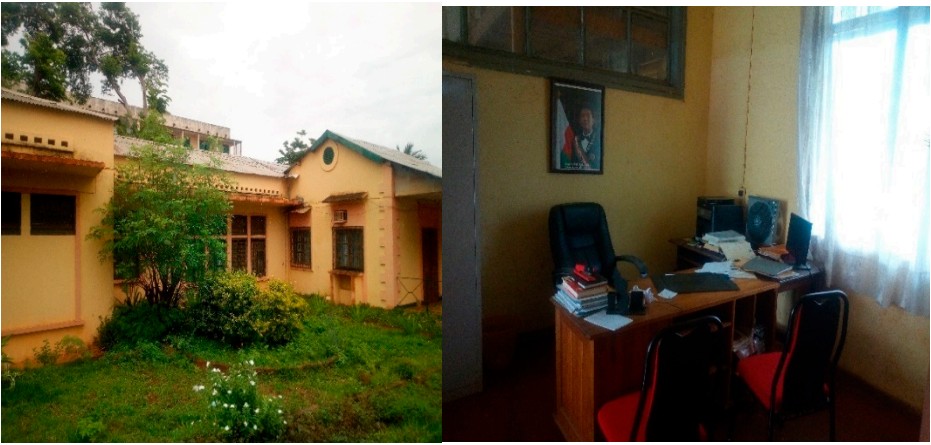

**Figure 2.** East facade of the studied office building.

The entire building was built on an area of 288 m$^2$ and height of 3.5 m. This building comprises 12 offices with 20 workers and one meeting room with a capacity of 50 people and two tool storage stores. The office occupancy schedule is from Monday to Friday, 8:00–12:00 and 2:00 pm–5:00 pm. To assess indoor air quality in more natural conditions, certain parameters were previously defined for this purpose (stop Heating, ventilation, and air conditioning (HVAC) system, Lighting: LED; activity of occupant...). The activity of the occupants can be considered as sedentary, that is, the workers were not subjected to any physical constraint. An estimation of the resistance of workers' clothes was around 1 Clo in the rainy season and 0.5 Clo in the dry season. In addition, the offices were analyzed under mechanical and natural ventilation conditions. Finally, the different materials making up this building and their characteristics are presented in Table 2. Air conditioning system: Fan Coil Unit (4-pipe) with seasonal cooling system (COP (Coefficient of Performance) = 1.8).

**Table 2.** Thermal properties of the building materials.

| Material | Building Element | Layer | Component | Thickness (m) | Thermal Conductivity (W/m-K) | Density (kg/m$^3$) | Specific Heat Capacity (J/kg K) | Embodied Carbon (kg$_{CO2}$) | U-Value (W/m$^2$-K) |
|---|---|---|---|---|---|---|---|---|---|
| Office | Exterior wall | Layer1 | Mortar | 0.02 | 0.5 | 1300 | 1000 | 0.12 | 2.62 |
| | | Layer2 | Concrete block | 0.15 | 1.63 | 2300 | 1000 | 0.08 | |
| | | Layer3 | Mortar | 0.02 | 0.5 | 1300 | 1000 | 0.12 | |
| | Partition wall | Layer1 | Mortar | 0.02 | 0.5 | 1300 | 1000 | 0.12 | 2.62 |
| | | Layer2 | Concrete block | 0.15 | 1.63 | 2300 | 1000 | 0.08 | |
| | | Layer3 | Mortar | 0.02 | 0.5 | 1300 | 1000 | 0.12 | |
| | Roof | Layer1 | Aluminum + 20% steel | 0.03 | 113 | 7000 | 390 | 3.31 | 2.01 |
| | | Layer2 | Ceiling | 0.02 | 0.056 | 380 | 1000 | 1.2 | |

The multi-zone model was created by the simulation tool. It was possible to have the data for each zone. However, the data that interests us are those of the entire building studied. The schedule of occupants was detailed in Section 2.3: The office occupancy schedule is from Monday to Friday, 8:00–12:00 and 2:00 pm–5:00 pm.

Other internal heat gains:

(a) Computer: Power density 1.1 W/m$^2$, radiant fraction 0.2; schedule: Work day from Monday to Friday, 8:00–12:00 and 2:00 pm–5:00 pm.
(b) Office equipment: Power density 0.8 W/m$^2$, radiant fraction 0.2.

*2.4. Simulation*

The design, modelling, and optimization of this building were carried out using the Design-Builder software. The Design-Builder software makes it very easy to study the energy performance of a building on the basis of its coupling with the Energy Plus software. Overall, this software has various similar functions such as modeling, optimization, analysis of environmental impacts, cost analysis, etc. All the equations relating to heat transfer and to the analysis of the cost of the building are directly incorporated into this software. The calculation time varies depending on the area of the building and the power of the simulator. In this sense, the modelling of the building studied by the Design Builder software is shown in Figure 3.

For a more reliable calculation, the different input data employed in the software were defined on the basis of the investigations previously carried out in this building. In this sense, the simulation interval chosen ranged from 1 January to 31 December with a time step of 2 min, and the infiltration rate was set around 0.7 ACH based on the ASHRAE recommendation for an average room [26].

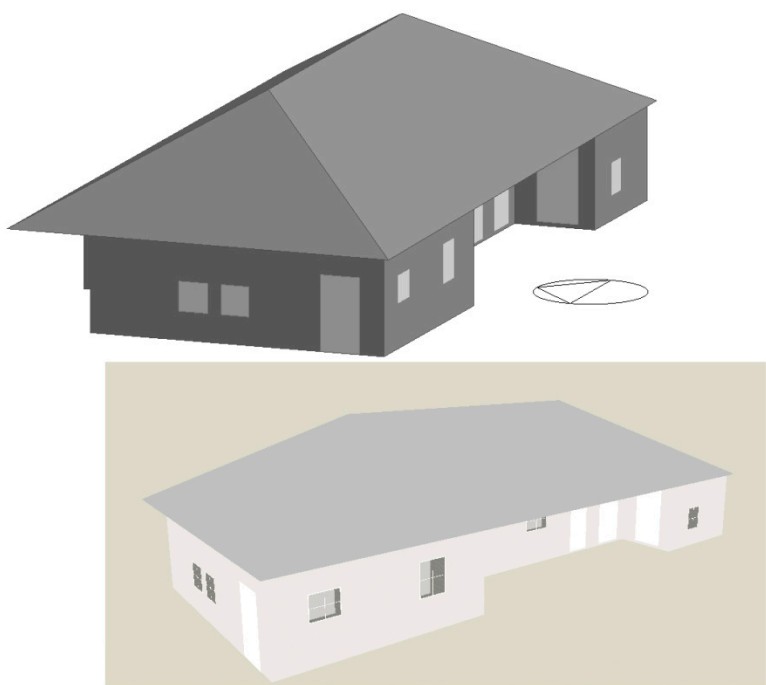

**Figure 3.** Building modeled with Design Builder tool.

At the same time, the recommended thermal comfort temperature range in the warm tropical region in Madagascaris between 22 °C and 28 °C all year round [24]. The range of 19–25 °C has been proposed for human comfort in sub-Saharan Africa in humid and hot tropical climate [25]. By consequence, in this research work, two temperature set points were set for the indoor air: 20 °C in cold months and 25 °C in hot months. Finally, the different wall structures studied are shown in Figure 4. Several scenarios were applied, as described in Section 2.7, and each applied test is shown in Figure 4.

## 2.5. Experiment

The experimental study was conducted in 2017, and also in 2019, in 25 municipalities located in several regions of Madagascar. As referenced in [27], in this study, 50 shopping centers, 5 hospitals, 67 traditional buildings, 25 schools, restaurants, offices, and hotels were investigated. The results were analyzed, compared, and integrated during two seasons (dry and rainy seasons). This study was conducted by 25 students. During this campaign, more than 1092 people were interviewed. The questionnaires were written in French and Malagasy, which are the two official languages of the country. More detailed information on the experimental study is shown in [3,27]. The comparison between the different measured and simulated data is detailed in Section 2.6, which calculates the different errors. The different physical measurements were taken at a height of 1.2 m in each office representing each area, as recommended by international standards. During the experimental study, each office was entitled to only one measurement point. Indeed, the measurement point was considered to be the place closest to the high concentration of office workers. The average temperature validated throughout the building, as shown in Figure 5, was that collected in all of the measurement points.

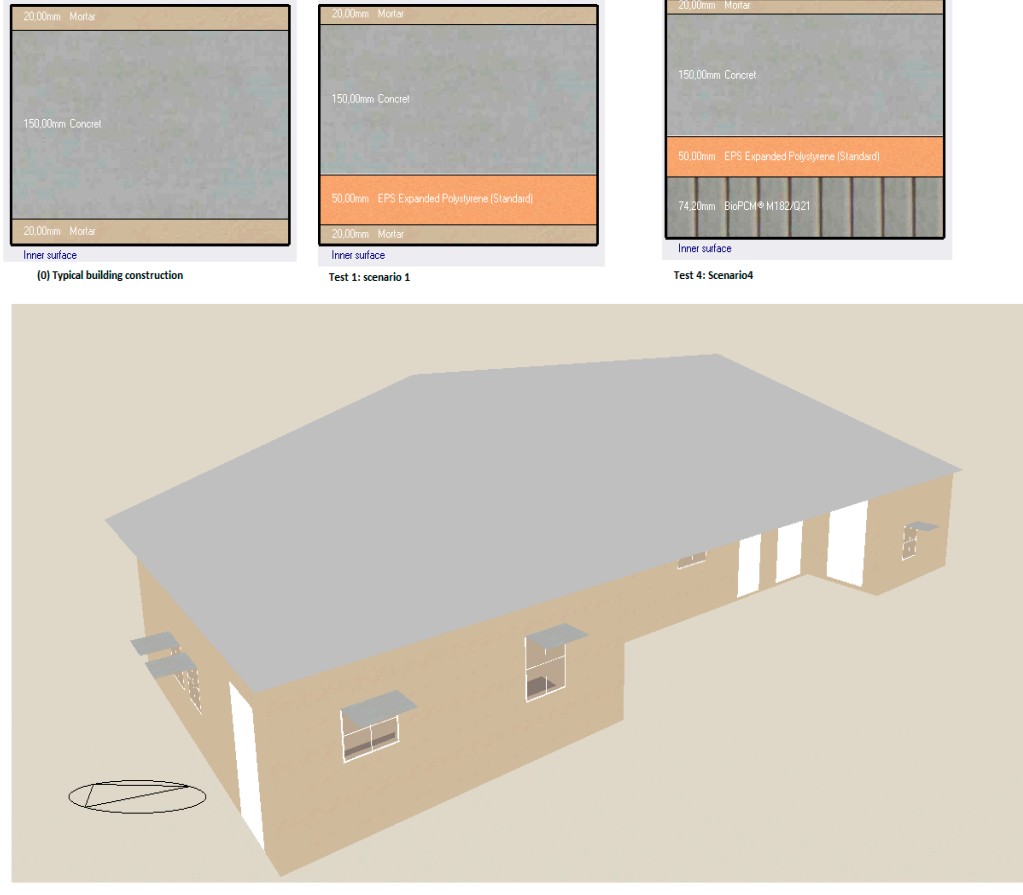

**Figure 4.** Typical uninsulated (0) and different scenarios.

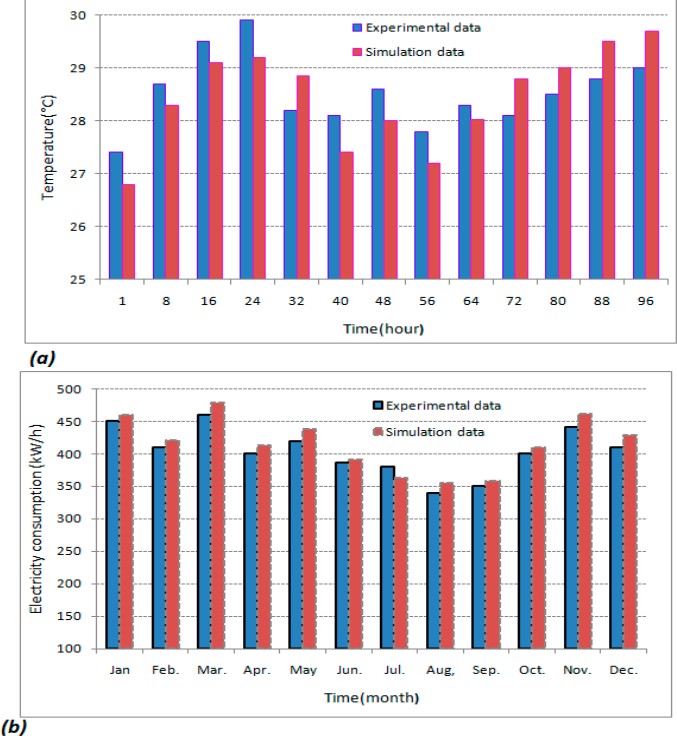

**Figure 5.** (**a**,**b**) Experimental and simulation data.

*2.6. Numerical Model Validation*

In a simulation study, the validation of the simulation model was the first step of the study to be conducted, one of the most important stages of the work. It verifies the veracity of the results. In the literature, one can find several methods allowing a new model of the simulation to be validated. In our case study, to compare the measured and simulated data, the coefficient of variation or the square root error (RMSE) and the mean bias error (MBE) were calculated, as it is indicated by ASHRAE in its guide14 [28]. In this report, it is indicated that a simulation model can be identified as validated if it meets the following conditions:

- Hourly MBE between ±10% and hourly RMSE less than 30%.
- Monthly MBE between ±5% and monthly RMSE less than 15%.

The different equations used as a basis for error calculations are mentioned in the ASHRAE guide. In this study, hourly temperature and monthly electricity data were used as a sample obtaining an MBE value of 8% and an RMSE value of 5%, where, once compared to the values suggested in the ASHRAE14 directive [28] of ±10% (case of MBE) and ±30% (case of RMSE), it can be concluded that our simulation is within the range recommended by ASHRAE. In this case, the model used in this research is considered to be validated on the basis of hourly temperature data. Experimental data were collected in 2019.

In addition, it is also necessary to compare the monthly energy values simulated and measured in these offices. In this sense, the MBE and RMSE values were −4% and around 7%, respectively. In the ASHRAE 14 guideline, the recommended monthly MBE limit was between ±5% and, for RMSE, between ±15%. By consequence, our found values lie within these intervals. This also means that the new model can be validated with the monthly energy data. Some experimental and simulation data are shown in Figure 5.

In summary, it was concluded that the simulation data was validated with hourly temperature and monthly values of electricity and, now, the new simulation model used in this study was validated.

*2.7. Forecasting*

Two Intergovernmental Panel on Climate Change (IPCC) [2] scenarios were used in this research to assess the evolution of climate and energy consumption: The low-impact scenario B1 and the high-impact scenario A2. Indeed, the A2 scenarios are of a more divided world. The family of A2 scenarios is characterized by a world of autonomous and independent functioning nations and a population in constant increase [29], while scenario B1 is of a more integrated and more ecological world. Hourly external data for the last 30 years of temperature, humidity, radiation, etc. are required for different projections.

*2.8. Scenarios*

Four simulation scenarios were obtained when combining the different constructive characteristics of the office buildings, but always considering their normal state and the input data showed before.

Scenario 1: The structure of the walls and the roof were changed by adding a 5 cm layer of expanded polystyrene.

Scenario 2: The insulation layer on the walls and the roof was removed and solar protections on all the windows were added to limit the penetration of solar irradiation.

Scenario 3: The insulation (as described in scenario 1) and shading of the building (as described in scenario 2) were combined.

Scenario 4: The insulation, with a shading effect, was simulated with the addition of a 7 cm phase change material layer (PCM), as shown in Figure 4.

## 3. Results and Discussion

This section is divided into five subsections: 3.1 Current performance of building, 3.2 Electricity consumption, 3.3 Cooling energy saving, 3.4 Net energy saving, and 3.5 Operational carbon.

### 3.1. Current Performance of Building

The different characteristics of office buildings in the four selected cities are detailed in Table 3, and in the case of natural conditions, Equations (1)–(3) give the optimal comfort temperature, and the upper and lower acceptability limit.

- Optimal comfort temperature (°C) [30,31]:

$$T_{ac} = 0.31 f(T_o) + 17.8 \ 10°C \le f(T_O) \le 33.5°C. \tag{1}$$

- Upper 90% acceptability limit is

$$T_{ac} = 0.31 f(T_o) + 20.3 \ 10°C \le f(T_O) \le 33.5°C. \tag{2}$$

- Lower 90% acceptability limit is

$$T_{ac} = 0.31 f(T_o) + 15.3 \ 10°C \le f(T_O) \le 33.5°C, \tag{3}$$

where $f(T_o)$ is the prevailing mean outdoor air temperature in ASHRAE 55 for 2013 and 2017.

**Table 3.** Some characteristics of indoor building performance.

| Cities | Antsiranana | Mahajunga | Toamasina | Taolagnaro |
|---|---|---|---|---|
| Operative temperature (°C) | 25–27 | 25–28 | 23–26 | 22–27 |
| Relative humidity (%) | 57–72 | 57–74 | 72–76 | 64–71 |
| Discomfort rate (%) | 17.8 | 19.4 | 34.5 | 25.9 |
| Cooling energy (kwh/m$^2$) | 77.5 | 81.9 | 50.6 | 48.7 |
| Energy consumption (kwh/m$^2$) | 173 | 186.7 | 130.4 | 127.1 |
| Operational carbon (kg$_{CO2}$/m$^2$) | 61.9 | 73.5 | 50.5 | 54.5 |

In Table 3, it can be observed that, in natural ventilation, the operative temperature is between 22 °C and 28 °C and the relative humidity is between 57% and 76% in the four cities. This comfort range is recommended by ASHRAE55, which fixed the operative temperature between 23 °C and 26 °C and relative humidity between 30% and 60% [32]. Despite this, more than 80% of workers find their workplace comfortable. These results are very interesting. They show that over the comfort range requested, by several international standards, the occupants found their environment comfortable. In general, among the four cities studied, indoor air is less comfortable and more polluting in Mahajunga and more comfortable in Toamasina. It was concluded that the climate is harsher in the city of Mahajunga, which is located in the zone of hot tropical climate.

Some detailed results of indoor air are shown in Table 4. The thermal comfort was analyzed during the hours of occupation from 8 am to 5 pm.

**Table 4.** Average indoor adaptive comfort in the different selected cities with 90% acceptability Range.

| | | No Strategy | | 2050 | | With Passive Strategies | |
|---|---|---|---|---|---|---|---|
| **Cities** | **Parameters** | **Min.** | **Max.** | **Min.** | **Max.** | **Min.** | **Max.** |
| Antsiranana | Temperature (°C) | 23.1 | 28.5 | 23.5 | 29.0 | 23.0 | 28.2 |
| | Relative humidity (%) | 56.0 | 73.0 | 43.1 | 52.2 | 58.6 | 74.8 |
| Mahajunga | Temperature (°C) | 23.2 | 28.6 | 23.4 | 29.0 | 23.0 | 28.2 |
| | Relative humidity (%) | 55.9 | 74.8 | 42.3 | 46.7 | 56.9 | 77.4 |
| Taolagnaro | Temperature (°C) | 22.3 | 28.5 | 22.7 | 29.0 | 22.0 | 28.1 |
| | Relative humidity (%) | 51.6 | 71.6 | 43.5 | 66.7 | 65.3 | 73.1 |
| Toamasina | Temperature (°C) | 22.5 | 28.4 | 22.8 | 29.0 | 22.0 | 28.2 |
| | Relative humidity (%) | 72.5 | 76.8 | 44.3 | 60.9 | 72.9 | 77.4 |

In this table, we can see that the passive strategy allows the improvement in indoor air (relative humidity and air temperature) between 10% and 25%, according to the cities and seasons. These results support the conclusions of several researchers who demonstrated that the application of passive strategy techniques can have a significant effect on the indoor climate [33,34]. Between 2017 and 2050, it is predicted that the indoor temperature will increase up to 1 °C, while the relative humidity will tend to decrease up to 5%. This is in agreement with the IPCC [35], which predicts that by 2050, if nothing is done to prevent the evolution of the weather, the weather temperature will increase to between 1.5 °C and 3.5 °C, depending on the regions and projection scenarios. Furthermore, the impact of this climate change will be visible with the high demand for energy in buildings [36]. In this sense, in the next subsection, we study the effects of PCM materials on the cooling energy consumption.

*3.2. Cooling Energy Saving*

Figure 6 shows the cooling energy saving after introducing the PCM material in the case of four cities. Globally, when we apply between 3 and 7 cm of the thickness of a PCM on the walls of an office building in coastal regions, the cooling energy saving is between 10% and 30%, depending on the region. Indeed, the cooling energy saved is estimated to be about 12.2% in Antsiranana, 12.5% in Mahajunga, 11.8% in Toamasina, and 11.9% in Taolagnaro during dry and rainy seasons. These results show that PCM materials have a more significant effect regarding cooling energy saving in warm tropical climates (e.g., Mahajunga city) than other coastal climate regions.

In the four cities, the energy saving is very low in the dry season (May–Oct.) and estimated as 18%–37%. In view of all these results, it is interesting to deduce that cooling energy is one of the main components of energy consumption in hot zones. According to EIA [37], globally, the electricity consumed in the residential sector for cooling was estimated at 214 billion kWh in 2018, or about 15% of the total electricity consumption of the residential sector in the world.

In the next subsection, the effect of different passive strategies on the total electricity consumption in the office will be evaluated.

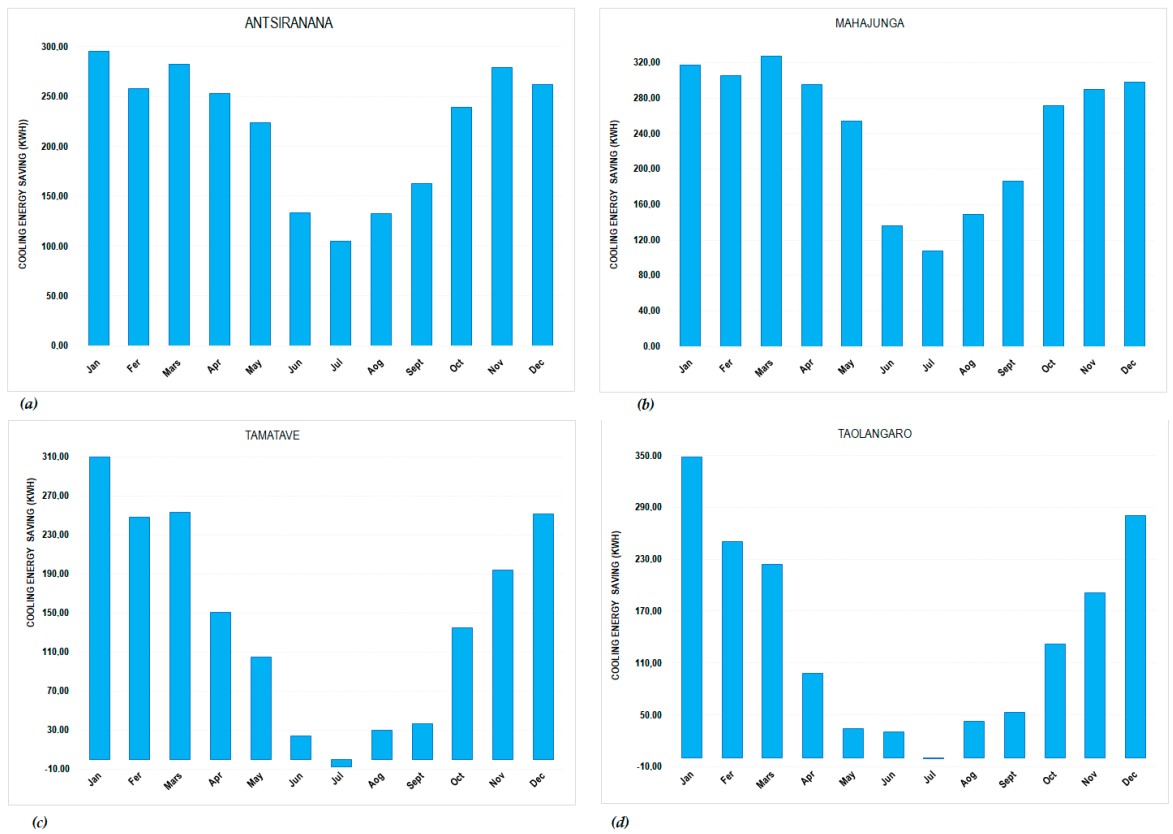

**Figure 6.** Cooling energy saving in the four cities during the two seasons.

### 3.3. Electricity Consumption

Figure 7 shows the annual quantity of electricity consumed for cooling, lighting, etc., in this office building when it is built in the four cities. The main results showed an estimated annual electricity consumption of between 88.2 and 121.3 kWh/m$^2$ in the four cities, and this amount is estimated to between 81.7 and 110.7 kWh/m$^2$ after setting 5 cm of insulation on the walls and roof, between 79.0 and 107.8 kWh/m$^2$ after setting 7.4 cm of PCM material, and between 74.1 and 101.6 kWh/m$^2$ after combining insulation and building shading. At the same time, an electricity consumption between 30% and 40% bigger in the hot tropical coastal (e.g., Mahajunga city) than in the humid tropical coastal region was observed (Toalagnaro city). The distribution of electricity consumption per strategy is shown in Table 5. The PCM has a very significant effect on the indoor climate. Indeed, Sovetova et al. [9] showed that optimal PCMs are capable of reducing temperature fluctuations and the maximum temperature up to 2.04 °C.

**Table 5.** Electricity consumption (kWh/m$^2$).

| Cities | No Strategy | With Shading Effect | With Insulation | With PCM | With Insulation + Shading | 2050 |
|--------|-------------|---------------------|-----------------|----------|---------------------------|------|
| Antsiranana | 115.6 | 111.8 | 104.2 | 101.6 | 95.7 | - |
| Mahajunga | 121.4 | 119.2 | 110.7 | 107.9 | 101.6 | 129.3 |
| Taolagnaro | 88.2 | 87.3 | 81.6 | 79.1 | 74.1 | 90.3 |
| Toamasina | 90.1 | 85.5 | 83.0 | 80.7 | 75.6 | - |

From this table, it can be concluded that the consumption of electrical energy varies according to the seasons. This rate remains very low compared to that of developed countries. Globally, it can be observed that the energy demand is expected to increase between 10% and 25% in 2050, and up to 85% in 2100, when the air temperature increase is between 1.5 °C and 3.5 °C. Wang et al. [38] showed that

the energy demand is expected to increase between 120% and 530% with an increase in air temperature between 2 °C and 5 °C in the next decade in Australia. Some results found in these results confirm these findings.

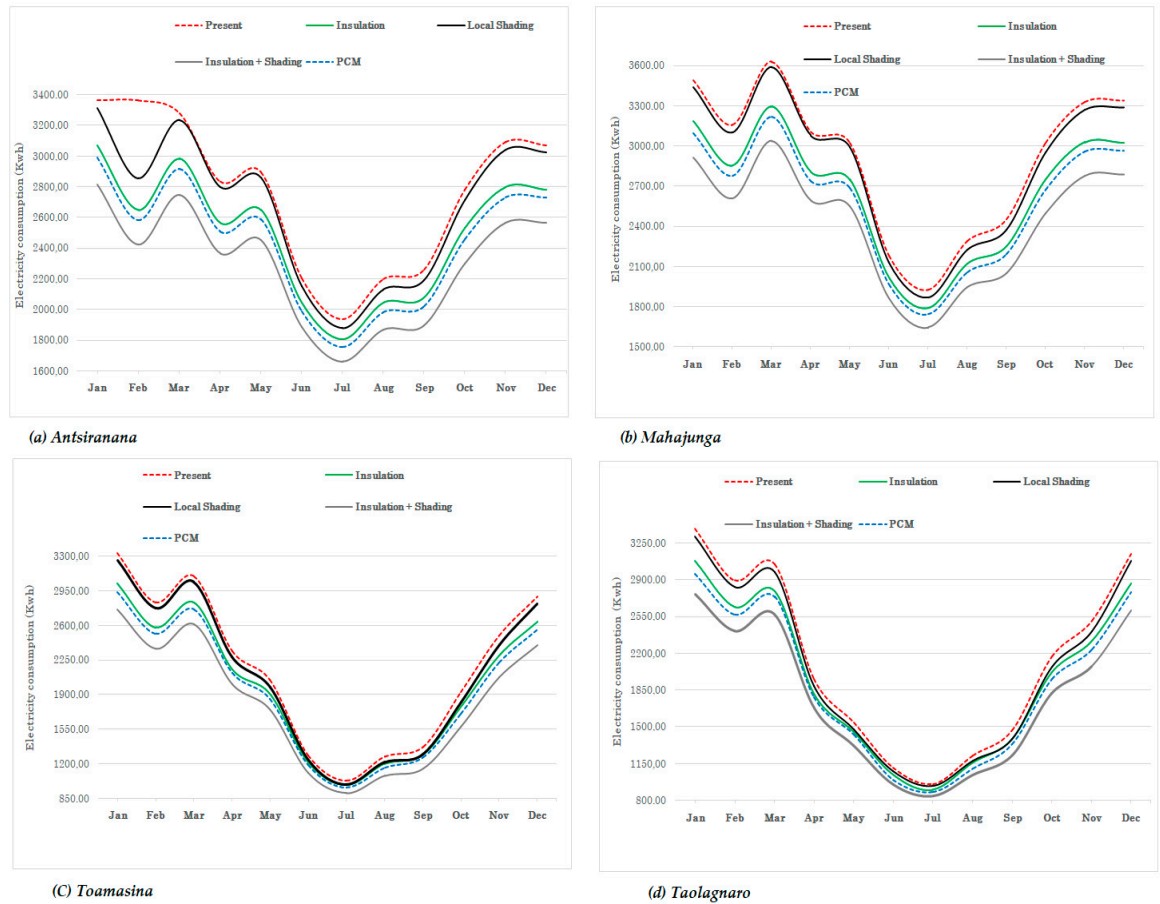

**Figure 7.** Variation in electricity consumption in the four cities selected after applying passive strategies.

### 3.4. Net Energy Saving

Figure 8 shows an energy reduction amount according to the four passive strategies proposed. Overall, in the four cities located in the coastal zone, the application of insulation on the walls and roof allows energy consumption to be reduced by 8%–15%; the introduction of PCM equipment reduced between 10%–18% of total energy per year, and finally, by combining insulation, shading, and PCM material, a reduction in the energy consumption is estimated by 15%–35%.These results also show that after 2050, passive strategies will be insufficient to improve the thermal performance of the building, due to the strong pressure of global warming. These results confirm the conclusion of Wang and Chen [39] in the case of San Diego city.

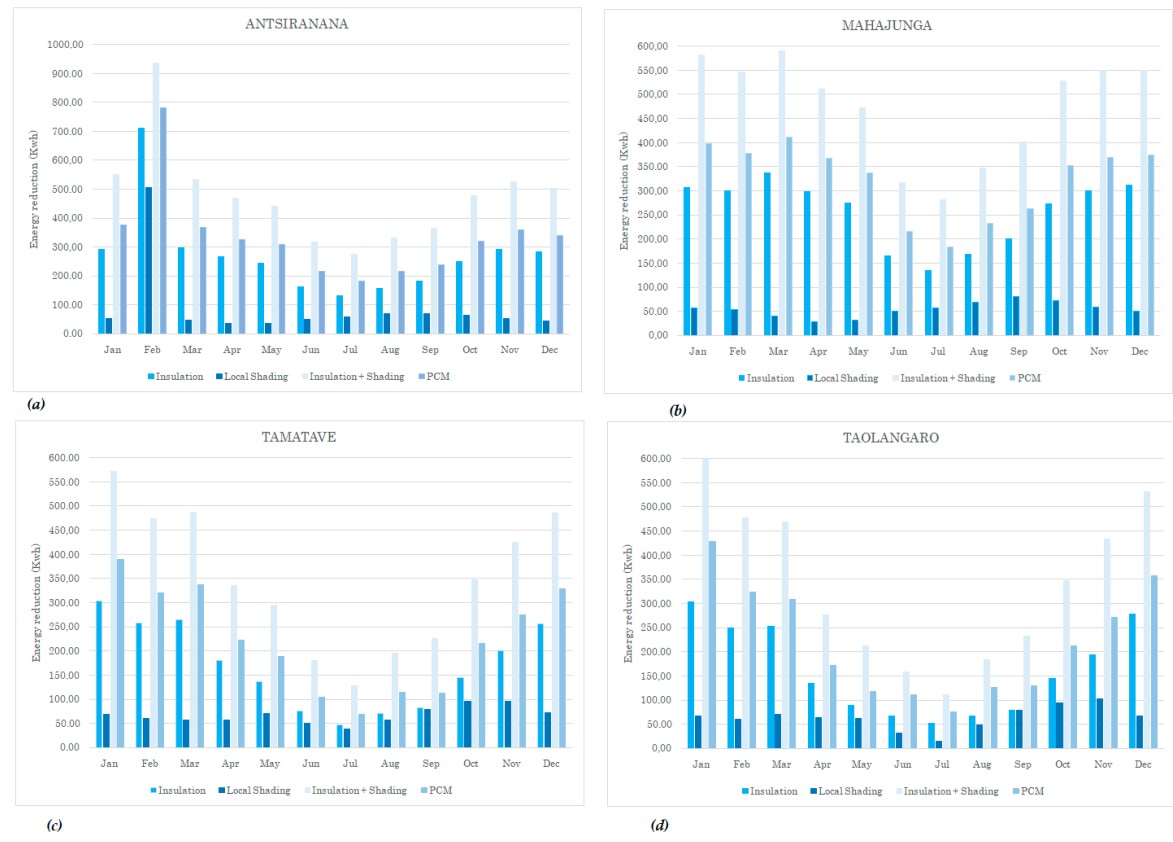

**Figure 8.** Energy reduction in the four cities with application of passive strategies during two seasons.

The passive strategy allows savings of 30%–40% of energy consumption in warm tropical climates compared to the humid tropical climate. These findings show that the energy yield of the passive strategy technique is the most important in the warm tropical climate. The results found in this research confirm certain conclusions found by Ahangari and Maerefat [40], who found that PCM materials make it possible to improve the conditions of thermal comfort and to minimize the energy demand in buildings.

*3.5. Operational Carbon*

Passive strategies can have a significant effect on the $CO_2$ emission, as shown in Figure 9 and Table 6. It is very interesting to notice that the $CO_2$ concentration decreases between 5% and 10% with the application of passive strategies methods, and it is expected to increase between 10% and 15% from 2017 to 2050.

**Table 6.** Operational carbon in the four cities.

| | | No Strategy | | With Passive Strategies | | 2050 | |
|---|---|---|---|---|---|---|---|
| **Cities** | **Parameters** | **Min.** | **Max.** | **Min.** | **Max.** | **Min.** | **Max.** |
| Antsiranana | Carbon $(kgCO_2/m^2)$ | 50.6 | 71.0 | 47.8 | 67.0 | 59.0 | 79.0 |
| Mahajunga | Carbon $(kgCO_2/m^2)$ | 56.2 | 76.2 | 51.5 | 71.6 | 63.5 | 83.5 |
| Taolagnaro | Carbon $(kgCO_2/m^2)$ | 38.7 | 58.7 | 34.8 | 54.9 | 43.4 | 63.4 |
| Toamasina | Carbon $(kgCO_2/m^2)$ | 38.8 | 58.9 | 35.8 | 56.0 | 44.6 | 64.6 |

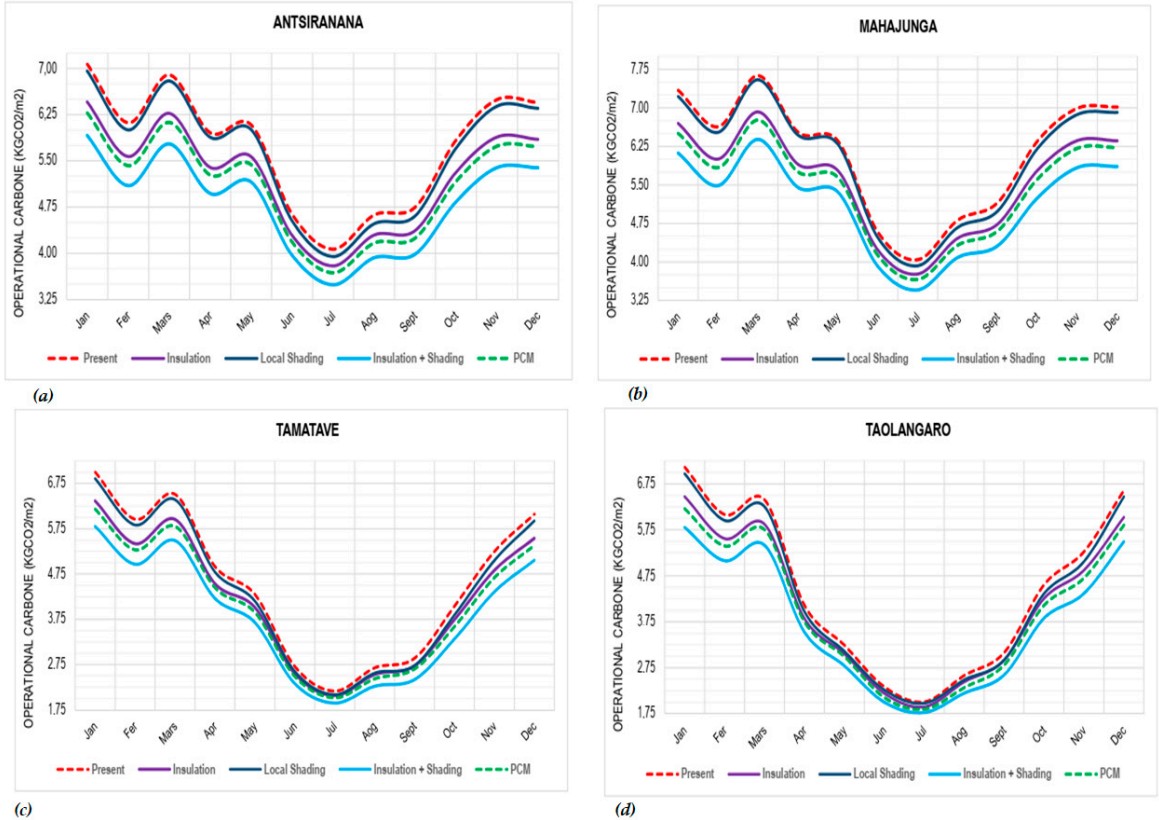

**Figure 9.** Carbon dioxide produced during the operational phase of office buildings in four cities.

Mahajanga city showed the highest concentration of $CO_2$ while Taolagnaro city showed the lowest. It was noticed in several studies that the concentration of $CO_2$ converges in proportion to the energy demand. This effect is not random; in fact, the sources of electricity production generate $CO_2$ when they are not renewable.

## 4. Statistical Analysis

Statistical analysis allows us to understand, in depth, whether the application of passive strategies significantly affects the performance of the building in coastal tropical regions. In this study, we will use four key parameters in the analysis: Air temperature, relative humidity, $CO_2$, and electricity consumption. The analysis is made by the recognized SPSS statistical software in this field.

In this research, to study the different details in depth, two statistical tests were applied: The *t*-test and chi-square carried out with a confidence level (CL) of 95%. The results of the *t*-test are reported in Table 7.

The main objective of chi-square is to compare two samples and determine the possible similarity or divergence between them. The significance interval is proposed and freely chosen with the SPSS software. Indeed, this is called the *p*-value (observed probability of a *t*-test).

The correlation is significant at *p*-value < 0.01.

In Table 7, a deep statistical analysis is shown by comparing four parameters of offices: Monthly air temperature, monthly relative humidity, $CO_2$, and electricity. The main results showed in the different cases have *p*-value < 0.01, which means that the two environments studied do not have the same performance. Therefore, it can be concluded that the application of passive strategies on the walls of office buildings (PCM materials and insulation) increases the performance of the environment. Finally, it was proposed to apply a chi-square test to verify these results, as shown in Table 8.

Table 7. Statistical analysis of different parameters.

| Datasets | Offices without Passive Strategy | Offices with Passive Strategy |
|---|---|---|
| **Air temperature (°C)** | | |
| Sample | 12 | 12 |
| Mean | 25.92 | 25.61 |
| Standard deviation | 0.42 | 0.31 |
| Confidence interval of the difference at 95% | [13.6; 14.2] | [13.4; 13.8] |
| Mean standard error | 0.12 | 0.09 |
| Correlation ($R^2$) | 0.99 | |
| *t*-test-results | 5.49 | |
| Degree of freedom | 11 | |
| *p*-Value | 0.0001 | |
| **Relative humidity (%)** | | |
| Sample | 12 | 12 |
| Mean | 63.49 | 65.24 |
| Standard deviation | 5.53 | 6.12 |
| Confidence interval of the difference at 95% | [47.9; 55.0] | [49.34; 57.1] |
| Mean standard error | 1.59 | 1.76 |
| Correlation ($R^2$) | 0.98 | |
| *t*-test-results | −8.76 | |
| Degree of freedom | 11 | |
| *p*-Value | 0.0002 | |
| **$CO_2$ emissions (kg)** | | |
| Sample | 12 | 12 |
| Mean | 1657.69 | 1391.34 |
| Standard deviation | 284.95 | 229.53 |
| Confidence interval of the difference at 95% | [1476.6; 1838.7] | [1245.5; 1537.2] |
| Mean standard error | 82.26 | 66.26 |
| Correlation ($R^2$) | 0.99 | |
| *t*-test-results | 16.28 | |
| Degree of freedom | 11 | |
| *p*-Value | 0.0001 | |
| **Electricity (kWh)** | | |
| Sample | 12 | 12 |
| Mean | 2735.48 | 2295.95 |
| Standard deviation | 470.23 | 378.76 |
| Confidence interval of the difference at 95% | [2436.7; 3034.3] | [2055.3; 2536.9] |
| Mean standard error | 135.74 | 109.34 |
| Correlation ($R^2$) | 0.99 | |
| *t*-test-results | 16.28 | |
| Degree of freedom | 11 | |
| *p*-Value | 0.0001 | |

**Table 8.** Chi-square tests for the offices without passive strategy and offices with passive strategy.

| Confidence Level (99%) | Value | df | Asymp. Sig. (2-sided) |
|---|---|---|---|
| Air temperature | | | |
| Pearson Chi-Square | 132.000 [a] | 121 | 0.023 |
| Likelihood Ratio | 59.63 | 121 | 1.00 |
| Linear-by-Linear Association | 10.94 | 1 | 0.001 |
| N-observations | | 12 | |

Superscript (a) indicates that it is a RC table chi-square test, and a correlation is significant at $p < 0.05$, 2-tailed level.

From Table 8, it can be deduced that office environments showed differences (*p*-value equal to 0.023, (0.023 < 0.05)), and it justifies the performance of environmental change with the introduction of PCM materials and insulation of the walls. What is more, these two analyses show that the performance of the building increases with the implementation of passive strategies.

The coastal zones of Madagascar are known in the world as places strongly dominated by the effects of global warming. This is not surprising; in fact, Madagascar is classified as the fourth country in the world most vulnerable to climate change [41,42]. In this sense, the strong current growth of the outdoor climate in this country has a significant effect on energy demand and indoor comfort. By consequence, the main results showed that one of the adaptive methods in this region will be to use the techniques of passive strategies.

## 5. Limitations of the Study

All scientific research has its limits. In the case of this study, it is seen that it was conducted only in an administrative office.

At the same time, the experimental study did not take place over several years, and it is sometimes recommended to carry out experimental studies over several years to validate the results of a new model. In addition, not all parameters describing thermal comfort have been studied.

The comparison is very weak because there is little work concerning the study of thermal performance of a building in the coastal zone. However, despite these limitations, these results can already be the subject of a new publication. What is more, this study constitutes a basis for certain specialists in the field who would like to improve the performance of buildings in various sectors.

## 6. Global Implication

The results found in this study can be extended to all coastal areas of the world located in tropical regions. In particular, the climatic data for 2050 was obtained on the basis of the hourly data for the last 30 years, and the main results can be of interest for designers to create more suitable buildings in the coastal areas. At the same time, the obtained results could also be very beneficial to investors, tourists, and politicians. Finally, it is expected that future research works and standards may arise from the results of this study like, for instance, the implementation of a construction standard more suited to coastal regions.

## 7. Conclusions

In conclusion, in this study, it is shown that the application of passive strategy techniques can be very beneficial as a strategy for adapting to the new climate and reducing the demand for energy in coastal zones. In fact, in the coastal zone, the implementation of passive strategies increases the comfort rate by between 10% and 25% while decreasing the cooling energy by 10%–30% depending on the region.

In all the countries of the world, coastal zones are often very popular. They are often considered to be regions with high temperature growth due to their geographic position. In particular, the results

of the statistical study showed a great difference between the case of normal office operating conditions and the case of application of passive strategies:

-    The addition of 50 mm thick expanded polystyrene on the different facades of the office walls reduces the average interior temperature up to 0.6 °C and saves up to 10% of total energy consumption in the coastal tropical region.
-    The external shading test reduces the average indoor temperature by up to 0.3 °C and saves up to 4% of total energy consumed in coastal tropical regions.
-    The addition of 74 mm of PCM thickness on the different facades of the office walls and roof allows the average indoor air temperature to be reduced by up to 0.5 °C and to save up to 19% of total energy consumption in coastal tropical regions.

It can be said that the results of the statistical analysis of the data revealed a strong performance of passive strategy buildings, which vary according to the geographical position of each city.

Globally, this research shows that passive strategies can be one of the best ways to mitigate the impacts of climate change in commercial buildings located in the coastal zone; they reduce energy consumption while increasing the thermal performance of the building. In particular, the choice of local materials more suited to the micro-climate during the design of office buildings allows discomfort and energy demand to be reduced in offices. Future research works will be oriented toward statistical analysis of the data.

**Author Contributions:** Conceptualization, M.K.N. and J.A.O.; Methodology, M.K.N.; Software, M.K.N.; Validation, M.K.N. and J.C.V.; Formal Analysis, M.K.N.; Investigation, J.C.V.; Resources, M.K.N. and J.A.O.; Data Curation, M.K.N.; Writing-Original Draft Preparation, M.K.N., J.C.V. and J.A.O.; Writing-Review & Editing, M.K.N., J.C.V.; Visualization, M.K.N.; Supervision, J.A.O. All authors have read and agreed to the published version of the manuscript.

**Funding:** This research received no external funding.

**Conflicts of Interest:** The authors declare no competing interests.

**Data Availability:** The data that support this research and other findings of this paper are available from the corresponding author upon reasonable request of each reader.

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
