# Peer review of "Energy Efficiency and Thermal Performance of Office Buildings Integrated with Passive Strategies in Coastal Regions of Humid and Hot Tropical Climates in Madagascar"

_applsci, doi:10.3390/app10072438_

Round 1

Reviewer 1 Report

The research investigates the effects of some efficiency measures on buildings located in the coastal regions of humidity and hot climates in Madagascar. While the subject in itself is interesting and falls within the scope of the journal, the article lacks crucial detailing of the experimental design and settings, as well as a clear presentation of the results. Additionally, the paper presents a lot of typing mistakes. Therefore, I am afraid I cannot recommend the article for publication in its current state without major and substantial rewriting of its contents.

There are several major issues that should be addressed in rewriting the article:

  • With respect to line 147, the considered parameters have to be listed
  • Figure 4b: before the figure, no information about the stratigraphy of the wall is given. This can confuse the reader. Additionally, the stratigraphy of the other three scenarios should be provided.
  • Section 2.5: the authors declare to have calibrated the simulation model, but no information about how and how long they acquired the experimental measurements are provided. In addition, the authors are recommended to show the comparison between experimental and simulation data.
  • Line 242-243: how was the workers' satisfaction evaluated?
  • Generally, to can adequately compare the result obtained for different scenarios, the results for each of them have to be presented, as reported in Table 5 (where, however, scenario 2 misses).

Author Response

Reviewer #1:The research investigates the effects of some efficiency measures on buildings located in the coastal regions of humidity and hot climates in Madagascar. While the subject in itself is interesting and falls within the scope of the journal, the article lacks crucial detailing of the experimental design and settings, as well as a clear presentation of the results. Additionally, the paper presents a lot of typing mistakes. Therefore, I am afraid I cannot recommend the article for publication in its current state without major and substantial rewriting of its contents.

Dear reviewer,

All the responses of all the comments were showed in blue colour in the revised manuscript.

 Comments
Responses
Comment1.

Extensive editing of English language and style required

Response1.

English style has been revised in all the manuscript

Comment 2.

With respect to line 147, the considered parameters have to be listed
Response 2.

-Stop of the HVAC system;
- Considered that the activity of workers is sedentary
- Fixed the resistance of clothing  at 1Clo  in rainy season and 0.5 Clo in dry season.

Comment 3.

Figure 4b: before the figure, no information about the stratigraphy of the wall is given. This can confuse the reader. Additionally, the stratigraphy of the other three scenarios should be provided.

Response3

All the scenarios are described on the figure4(scenario 3 = Scenario 2 + scenario 1).
Comment4.

Section 2.5: the authors declare to have calibrated the simulation model, but no information about how and how long they acquired the experimental measurements are provided. In addition, the authors are recommended to show the comparison between experimental and simulation data.
Response 4.

The experimental study was conducted in 2017, and also in 2019, in 25 municipalities located in several regions of Madagascar.

As referenced in [37], in this study, 50 shopping centres, 5 hospitals, 67 traditional buildings, 25 schools, Restaurants, office and Hotels were investigated during two seasons. The results were analysed, compared and integrated. This study was conducted by 25 students. During this campaign, more than 1,092 people were interviewed. The questionnaires were written in French and Malagasy which are the two official languages of the country. More detailed information on the experimental study is shown in [3,37]. The comparison between the different measured and simulated data is detailed in section 2.6, which calculates the different errors.The comparison is showed based on figure5.

Comment 5.

Line 242-243: how was the workers' satisfaction evaluated?

Response5.

The workers' satisfaction was evaluated under basis of statistical analysis of 1039 questionnaire analysed during this study carried in the offices, schools, residence etc.

Please see reference:

Modeste Kameni Nematchoua, Paola Ricciardi, CinziaBuratti.Statistical analysis of indoor parameters an subjective responses of building occupants in a hot region of Indian ocean; a case of Madagascar island. Applied Energy 208 (2017) 1562–1575.

Comment 6.

 Generally, to can adequately compare the result obtained for different scenarios, the results for each of them have to be presented, as reported in Table 5 
Figure 6

Table 5 has been improved with the addition of scenario2.

Reviewer 2 Report

Thank You very much for the possibility to become familiar with an interesting article. It is very well-written and has an empirical character. The Authors should be appreciated for the research reliability and proper selection of source literature, although it could be wider. The strong points of this article are its layout and the clarity of the presented contents. The well-prepared methodical part should be appreciated. The Global implications section should be included in Section 6 Conclusions. In general, in my opinion, the conclusion section should be wider. The conclusions are too poor for such rich research material. The obtained results should be wider referred to the research findings of other authors.

The article must be prepared in accordance with the formal guidelines of Applied Sciences.

Author Response

Reviewer #2:

Thank You very much for the possibility to become familiar with an interesting article. It is very well-written and has an empirical character. The Authors should be appreciated for the research reliability and proper selection of source literature, although it could be wider. The strong points of this article are its layout and the clarity of the presented contents. The well-prepared methodical part should be appreciated.

Response to reviewer #2:

Dear reviewer,

All the responses of all the comments were showed in blue colour in the revised manuscript.

Comments

Responses

Comment1.

The Global implications section should be included in Section 6 Conclusions.

Response1.

Global implication: this research shows that passive strategies can be one of the best ways to mitigate the impacts of climate change in commercial buildings located in the coastal zone; they reduce energy consumption while increasing the thermal performance of the building.

Comments2

In general, in my opinion, the conclusion section should be wider. The conclusions are too poor for such rich research material.

Response2.

 Conclusion section has been improved

Comment3

The obtained results should be wider referred to the research findings of other authors.

Response3.

The results have been referred with other studies.

Comment4.

The article must be prepared in accordance with the formal guidelines of Applied Sciences.

Response4.

The formal guidelines are respected in the revised manuscript.

Reviewer 3 Report

The paper deals with energy efficiency of office buildings integrated with passive strategies in Madagascar. The issue is topical and worth to be investigated. However the paper has some limitations and needs revision. The numbering of sections is incorrect. Please check introduction section numbering and other numbers. The structure of paper should be briefly presented in the end of introduction. The novelty and input of paper should be highlighted in introduction. The paper lacks discussion of results and comparison with other studies. The policy implications of the study need to be developed. Limitations of the study should clearly addressed and future research guidelines should be provided. Conclusions should better structured. The paper needs major revision. Also some technical issues and spelling mistakes should be corrected during revision.

Author Response

Reviewer #3: The paper deals with energy efficiency of office buildings integrated with passive strategies in Madagascar.The issue is topical and worth to be investigated. However the paper has some limitations and needs revision.

Response to reviewer #3:

Dear reviewer,

All the responses of all the comments were showed in blue colour in the revised manuscript.

Comments

Responses

Comment1.

The numbering of sections is incorrect. Please check introduction section numbering and other numbers.

Response1

The numbering of sections has been revised

Comment2.

The structure of paper should be briefly presented in the end of introduction. The novelty and input of paper should be highlighted in introduction.

Response2.

Originality and novelty are detailed in introduction section in blue colour

Comment3.

The paper lacks discussion of results and comparison with other studies.

Response3.

In the revised manuscript, the discussion and comparison have been improved.

Comment4.

The policy implications of the study need to be developed

Response4.

The results found in this study can be extended to all coastal areas of the World located in tropical regions. In particular, the climatic data for 2050 was obtained on the basis of the hourly data for the last 30 years and main results can be of interest for designers to create more suitable buildings in the coastal areas. At the same time, the obtained results could also be very beneficial to investors, tourists and politicians. Finally, it is expected that future research works and standards may arise from the results of this study like, for instance, the implementation of a construction standard more suited to coastal regions.

Comment5.

Limitations of the study should clearly addressed and future research guidelines should be provided.

Response5.

All scientific research has its limits. In the case of this study, it is seen that this study was conducted only in an administrative office.

The experimental study did not take place over several years, it is sometimes recommended to carry out experimental studies over several years to validate the results of a new model.

In addition, not all parameters describing thermal comfort have been studied.

The comparison is very weak because there is little work concerning the study of thermal performance of building in the coastal zone. However, despite these limitations, these results can already be the subject of a new publication. This study constitutes a basis for certain specialists in the field who would like to improve the performance of buildings in various sectors.

Comment 6

Conclusions should better structured.

Response6

Conclusions section has been improved

Comment 7

The paper needs major revision. Also some technical issues and spelling mistakes should be corrected during revision.

Response 7

English style has been revised by a native English speaker.

Reviewer 4 Report

The article must be re-written in good English.

Many words are used or exchanged like calibration and validation.

The results seem to be redundant and the charts just shifted of some value for each intervention strategy.

The same for the monthly histogram final results.

It seems just a Master student exercise.

It does not reach the minimum requirements for scientific publication.

The novelty is missing.

Too many self-citation from the authors.

Author Response

Reviewer #4: Comments and Suggestions for Authors

Response to reviewer #4:

Dear reviewer,

All the responses of all the comments were showed in blue colour in the revised manuscript.

Comments

Responses

Comment 1

The article must be re-written in good English.

Response 1

English language has been improved in all the manuscript

Comment 2

Many words are used or exchanged like calibration and validation.

Response 2

We have in some cases used the synonyms of words so as not to have repetitions of the same words.

Comment 3

The results seem to be redundant and the charts just shifted of some value for each intervention strategy.

Response 3

The presentation of the results has been improved. The study relates to the application of several passive strategies, so it is normal that the results vary according to the passive strategies.

There are not similar results, in each section, we showed the case of each parameter:

3.1. Current performance of building; 3.2. Cooling energy saving; 3.3. Electricity consumption; 3.4. Net energy saving; 3.5. Operational carbon

Comment 4

The same for the monthly histogram final results.

Response 4

The last figure evaluated Energy reduction in the four cities with application of passive strategies during two seasons. Towards this section we tried to understand the depth effects of passive strategy on the energy consumption.

Comment 5

It seems just a Master student exercise.

Response 5

Dear professor,
It’s not a Master student exercise,
We have spent enough time doing this research.

Comment 6

It does not reach the minimum requirements for scientific publication.

The novelty is missing.

Response 6

This article has been improved. Originality and novelty are improved upon introduction.

Comment 7

Too many self-citation from the authors.

Response 7

Analyse and discussions are improved by new citation.

Reviewer 5 Report

The paper presents an analysis of the impact of thermal insulation materials and phase change materials on thermal comfort and energy demand in office buildings located in coastal areas in the hot and humid tropical climate in Madagascar.

In the article there are several parts that require clarification before this paper is accepted for the publication.

Section “Introduction”

In my opinion this section is well written. The latest literature has been cited. A research gap is indicated.

Section “Methodology”

Table 1. Enter the year from which these data are provided.

Lines 133 to 136. Are climate data generated for a specific calendar year or standard data, e.g. based on 30 years of observations?

Lines 150 to 151. In the later description of the model, only infiltration is mentioned.

Table 2 window data was omitted: U-value, SHGC-value. What about window shading?

Line 167 Adopting a constant value of the infiltration air flow is a very rough approximation. Has it been confirmed that the average value of this air change rate is 1 ACH? With natural ventilation, such a high value is very difficult to achieve.

Was the single-zone or multi-zone model built in Design Builder? There is no information about the schedule of occupants, which was adopted in the model. What about other internal heat gains, e.g. computers?

Lines 171 to 174 The  recommended  thermal  comfort  temperature  range  in  this  region  is between 22 °C and 28 °C and next 20oC was assumed? Why? Did the cooling system work around the clock, or only during occupant's hours?

Section "Calibration" is completely unclear. Measured and calculated temperatures are compared. In this case, you had to make measurements about which there is no information in the article. For what climate data was this validation done, for the real year? Validation of hourly temperature values was carried out for the whole year? Obtained error is very low, below 1%, this is strange considering the many rough assumptions about the model (e.g. ACH). Only the final error result is given, it is not known what parameters were changed during calibration. The results have not been supported by any figures, e.g. hourly variation of measured and calculated temperature.

Section “Results and Discussions”

The authors refer to ASHRE-55 when analyzing thermal comfort. This standard is based on operative temperature not air temperature, e.g. range 23 to 26 oC in line 241 is the operative temperature range. Throughout this analysis, the authors have mixed up the concepts of operative temperature and air temperature. Formulas in lines 230 to 235  specify operation temperature but in tables there are air temperature values.

In Table 3 energy consumption is presented. In that case, what about the cooling system efficiency used for calculations? What does “Average cooling energy (kWh/m2)” mean? Average temperature and average humidity: average of what? Average operational carbon: only for cooling or total for the whole building?

For what hours of the day was thermal comfort analyzed? Around the clock or only during occupant's hours?

Figure 5: A comma means a decimal mark? If so, it should be a full stop.

Section 3.5. What was the CO2 emission calculated from? Only for cooling or total for building (for building structure, lighting etc.)?

Section “Conclusions”

Conclusions are well written to the analysis presented. However, due to the required changes in methodology, you may need to improve them.

Author Response

Reviewer #5: The paper presents an analysis of the impact of thermal insulation materials and phase change materials on thermal comfort and energy demand in office buildings located in coastal areas in the hot and humid tropical climate in Madagascar.

In the article there are several parts that require clarification before this paper is accepted for the publication.

Response to reviewer #5:

Dear reviewer,

All the responses of all the comments were showed in blue colour in the revised manuscript.

Comments

Responses

Comment 1

Section “Introduction”

In my opinion this section is well written. The latest literature has been cited. A research gap is indicated.

Response1

Thanks

Comment 2

Table 1. Enter the year from which these data are provided.

Response 2

These data are provided in 2019.

Comment 3

Lines 133 to 136. Are climate data generated for a specific calendar year or standard data, e.g. based on 30 years of observations?

Response3

With this simulation tool, it is possible to have annual data for each of the regions of the globe. However to make projections in 2030 or 2050, we have used hourly data from the past 30 years as recommended by the IPCC.

Comment 4

Lines 150 to 151. In the later description of the model, only infiltration is mentioned.

Response4

The model is described in several paragraphs according to the developed subtitles, with the example of section2.3, which the description of the model has been improved.

Comment 5

Table 2 window data was omitted: U-value, SHGC-value. What about window shading?

Response 5

Some characteristics of window are given in the table 2.

Comment 6

Line 167 Adopting a constant value of the infiltration air flow is a very rough approximation. Has it been confirmed that the average value of this air change rate is 1 ACH? With natural ventilation, such a high value is very difficult to achieve.

Response 6

This value was 0.7 ACH, with natural ventilation

Comment 7

Was the single-zone or multi-zone model built in Design Builder? There is no information about the schedule of occupants, which was adopted in the model. What about other internal heat gains, e.g. computers?

Response 7

The multi-zone model was created by the simulation tool. It was possible to have the data for each zone. However, the data that interests us is that of the entire building studied. The schedule of occupants was detail in section 2.3: The office occupancy schedule is from Monday to Friday 8:00 -12: 00and 2:00 pm-5:00pm.

Other  internal heat gains:

(a) Computer:  power density 1.1W/m2, radiant fraction 0.2;schedule: work day from Monday to Friday.

(b)  Office equipment: power density 0.8 W/m2, radiant fraction 0.2.

Comment 8

Lines 171 to 174 The recommended thermal comfort temperature range in this region is between 22 °C and 28 °C and next 20oC was assumed? Why? Did the cooling system work around the clock, or only during occupant's hours?

Response 8

The thermal comfort temperature range is often different from the uniform or operational temperature range of the air inside the building.
Studies previously carried out in this region proposed a comfort range of 22 to 28 ° C; however, sometimes the internal temperature decrease up to 18 ° C.
In this study, the set for the indoor air:
20 °C in cold months.

Cooling system only work during occupant's hours: from Monday to Friday 8:00 -12: 00and 2:00 pm-5:00pm.

Comment 9

Section "Calibration" is completely unclear. Measured and calculated temperatures are compared. In this case, you had to make measurements about which there is no information in the article. For what climate data was this validation done, for the real year? Validation of hourly temperature values was carried out for the whole year? Obtained error is very low, below 1%, this is strange considering the many rough assumptions about the model (e.g. ACH). Only the final error result is given, it is not known what parameters were changed during calibration. The results have not been supported by any figures, e.g. hourly variation of measured and calculated temperature.

Result 9

Experimental data were collected in 2019.

The comparison of data is showed in the figure 5.

Comment 10

The authors refer to ASHRE-55 when analyzing thermal comfort. This standard is based on operative temperature not air temperature, e.g. range 23 to 26 oC in line 241 is the operative temperature range. Throughout this analysis, the authors have mixed up the concepts of operative temperature and air temperature. Formulas in lines 230 to 235specify operation temperature but in tables there are air temperature values.

Response10

In the table3, it’s rather “Operative temperature”. This mistake has been corrected.

Comment 11

In Table 3 energy consumption is presented. In that case, what about the cooling system efficiency used for calculations? What does “Average cooling energy (kWh/m2)” mean? Average temperature and average humidity: average of what? Average operational carbon: only for cooling or total for the whole building?

Responses 11

- The HVAC template used in this study is the Fan Coil Unit (4-pipe), Air Cooler Chillers with cooling system seasonal COP =1.8.

- The Design Builder software provides simulation results by zone and that of the building covering all zones. It is that of the building that results from all of the building zones that we have called: “Average cooling energy (kWh/m2)” mean. But we think it's better to just say "Cooling energy".

- Thisis the monthly data.

- Average operational carbon for the whole building.

NB. Please note by observing the rate of carbon emitted by type of material in table 2, that the materials chosen in this study are low carbon emissions.

Comment 12

For what hours of the day was thermal comfort analyzed? Around the clock or only

during occupant's hours?

Response 12

The thermal comfort was analysed during the hours of occupation from 8AM to 5PM.

Comment 13

Figure 5: A comma means a decimal mark? If so, it should be a full stop.

Response 13

This mistake was corrected.

Comment 14

Section 3.5. What was the CO2 emission calculated from? Only for cooling or total for building (for building structure, lighting etc.)?

Response 14

This is operational carbon, i.e. supply during the building's operating phase (embodied carbon + equipment + electricity)

Comment 15

Conclusions are well written to the analysis presented. However, due to the required changes in methodology, you may need to improve them.

Response 15

The conclusions section was improved.

Round 2

Reviewer 1 Report

The manuscript has been properly improved and now is ready for publication.

Author Response

Thanks

English style has been further improved

Reviewer 3 Report

The authors have corrected their manuscript and have addressed all my comments in corrected version of manuscript. The authors have also provided answers to my comments. The paper has been improved and can be published in current version.

Author Response

Thanks

Reviewer 4 Report

Calibration and validation are not synonymous.

The results are the same ones and they seem a repetition of the same curve with a small shift.

Unfortunately, I do not see a complete work and even an original contribution.

It seems to me just a case study. Where is the novelty?

Author Response

 Comments

 Responses

Comment1

Calibration and validation are not synonymous.

The results are the same ones and they seem a repetition of the same curve with a small shift.

Response1.

- It was a confusion between these two words (Calibration and validation)  .

  We believe that this section is  the validation of the new model.

Besides, we have improved section: 2.6. Numerical model validation

Comment2.

Unfortunately, I do not see a complete work and even an original contribution.

It seems to me just a case study. Where is the novelty?

Response2.

 Dear Professor,

Original and novelty are mentioned in the introduction section:

However, the following identified problems are still unsolved.

a) The energy efficiency and thermal performance of buildings incorporating thermal insulation and PCM materials, located in coastal regions with a hot and humid tropical climate, have not yet been studied.

b) The relationship between carbon reduction, energy savings and weather factors has not yet been assessed in the tropical coastal zones.

In all regions of the World, coastal zones are highly exposed to the harmful effects of climate change, so that it is very difficult to propose a common international standard relative to these regions. In fact, the different impacts vary from one region to another. The island of Madagascar is one of the regions of the World where biodiversity and buildings are strongly impacted by the seasonal variability of the climate. So far, no study or standard was established in the big island to improve the performance of residential and commercial buildings. The various passive strategies remain little known to the population of the island of Madagascar.

For the moment, no study exists in the literature aimed at improving the performance of workers in buildings located in the coastal zone in the 6 Indian Ocean countries.

Reviewer 5 Report

The quality of this manuscript has been improved, but  not all of my questions have been answered.

Several parts of the article still need to be clarified before this paper is accepted for the publication.

Previous comment 7 “Was the single-zone or multi-zone model built in Design Builder?”

Why is this information not added to the article?

Previous comment 8 The recommended thermal comfort temperature range in this region is between 22 °C and 28 °C and next 20oC was assumed? Why?”

The authors did not explain why they assumed a temperature outside the recommended range. One of the aims of the article is the analysis of thermal comfort, while the authors for the simulation assumed temperature outside the comfort zone. So in advance the result is doomed to failure.

Previous comment 9 Section "Calibration" is completely unclear. Measured and calculated temperatures are compared. In this case, you had to make measurements about which there is no information in the article. For what climate data was this validation done, for the real year? Validation of hourly temperature values was carried out for the whole year? Obtained error is very low, below 1%, this is strange considering the many rough assumptions about the model (e.g. ACH). Only the final error result is given, it is not known what parameters were changed during calibration. The results have not been supported by any figures, e.g. hourly variation of measured and calculated temperature.”

Unfortunately, this section has not been improved. Nothing is known about the measurements, except that they were carried out in 2019. The building consisted of several rooms. How many temperature measurement points were there (in one room, in selected rooms, in all rooms)? What was validated, average temperature in the whole building? What is on Figure 5, the average temperature for the building?

Figure 5 shows measured and calculated temperature for 13 days tested. It can be seen that the differences are definitely greater than 0.8%. Where does MBE = 0.8% and RMSE = 0.5% come from? In my opinion, something is wrong here.

In this section from lines 226 to 232 change „calibration” to validation”. The title of Section 2.6 rather should be “Validation”

It is still unknown which  parameters of buildings were changed during calibration. Was the model correct at the first step of validation?

Author Response

Reviewer #5.  Several parts of the article still need to be clarified before this paper is accepted for the publication.

Comments

Responses

Comment1.

Previous comment 7 “Was the single-zone or multi-zone model built in Design Builder?”

Why is this information not added to the article?

Response1.

 This information is added in the article, section 2.3 .

 Comment2.

Previous comment 8 “The recommended thermal comfort temperature range in this region is between 22 °C and 28 °C and next 20oC was assumed? Why?”

The authors did not explain why they assumed a temperature outside the recommended range. One of the aims of the article is the analysis of thermal comfort, while the authors for the simulation assumed temperature outside the comfort zone. So in advance the result is doomed to failure.

 Response2.

- The  thermal comfort temperature range in warm tropical region  is between 22 °C and 28 °C in Madagascar.

- The  thermal comfort temperature range in humid and warm tropical region in Sub-Saharan African   is between 19-25°C [42].

 In consequence, in this research work, two temperature set points were set for the indoor air: 20°C in cold months and 25 °C in hot months(see section 2.4).

Comment3.

Unfortunately, this section has not been improved. Nothing is known about the measurements, except that they were carried out in 2019. The building consisted of several rooms. How many temperature measurement points were there (in one room, in selected rooms, in all rooms)? What was validated, average temperature in the whole building? What is on Figure 5, the average temperature for the building?

Response3.

The different physical measurements were taken at a height of 1.2m in each office representing each area, as recommended by international standards. During the experimental study, each office was entitled to only one measurement point. Indeed, the measurement point was considered to be the place closest to the high concentration of office workers. The average temperature validated throughout the building as shown in Figure 5 was that collected in all of the measurement points.(see section 2.5)

Comment4.

Figure 5 shows measured and calculated temperature for 13 days tested. It can be seen that the differences are definitely greater than 0.8%. Where does MBE = 0.8% and RMSE = 0.5% come from? In my opinion, something is wrong here.

 Response4.

We verified the calculations again,

- the results give MBE = 08% and RMSE = 05%;

 we have    missed   previously results (see section 2.6)

Comment5.

In this section from lines 226 to 232 change „calibration” to validation”. The title of Section 2.6 rather should be “Validation”

Response5.

 The title  of section 2.6 has been changed

In this section from lines 226 to 232 change „calibration” to validation”. The title of Section 2.6 rather should be “Validation”

It is a validation of the simulation model.
We used two parameters for this purpose:
-the hourly temperature data
-Monthly electricity data.